# OpenReview forum: "On the Exploitability of Instruction Tuning"
_NeurIPS.cc/2023/Conference — NeurIPS 2023 poster_

### Official Review · Reviewer_oPzU · 2023-06-25

**Soundness:** 3 good
**Presentation:** 3 good
**Contribution:** 3 good
**Rating:** 6
**Confidence:** 5

**Summary:**

The authors propose AutoPoison, an automated data poisoning pipeline.
They demonstrate two types of attacks: content injection (e.g, brand names), and over-refusal attacks.
The authors demonstrate how AutoPoison can change a model’s behavior by poisoning only a small fraction of data while maintaining a high level of stealthiness in the poisoned examples.

**Strengths:**

1) The paper is well-written and easy to follow.
2) The idea is simple and intuitive, yet new at least in terms of the proposed attack pipeline.


**Weaknesses:**

1) I'm mainly missing comparisons on more recent models, such as MPT, Falcon, LLaMA, etc.
2) the evaluation protocol for the proposed over-refusal attack should be much broader in order to extract meaningful insights from it.
3) Can the authors succeed on open source oracles? without using OpenAI's GPT?

I now that some of the questions were mentioned in the limitations section, but in my opinion they are critical question that needs to be addressed.

**Questions:**

See the Weaknesses section.

**Limitations:**

authors addressed the limitations and, if applicable, potential negative societal impact of their work.

---

> ### Author Rebuttal · Authors · 2023-08-10
>
> > Evaluation of more recent models.
>
> Thank you for the suggestion. We are working on extending the evaluation to more recent models.  Poisoning experiments often require training a model many times on a range of poison ratios. The cost of using newer models is high since recent models like Llama do not have variants as small as OPT-1.3B. We are investigating parameter-efficient training strategies to fine-tune larger models. We may need more bandwidth to complete the experiments before the discussion period ends, but we are hoping to get a compute allocation large enough to include them in this paper's next version.
>
> > More comprehensive evaluation protocols.
>
> Thank you for this suggestion as well. As discussed in section 4.3, evaluating the over-refusal attack is not a straightforward problem. For this reason, we have broadened our evaluation protocol by using an LLM as the judge (compared to conventional metrics or rule-based evaluations). Nonetheless, given the tricky nature of this problem, we agree that there are limitations in our current protocol.
> Based on the attack goal of our threat model, our evaluation focuses on two specific aspects to check if the refusal message is valid (for effectiveness) and provides reasons (for stealthiness). Depending on the specific use cases, one might add additional dimensions to their evaluation.
>
>
> > AutoPoison with open-source oracles.
>
> This is an interesting point. Thank you for the thoughtful suggestion. We have now conducted additional experiments to verify the effectiveness of AutoPoison with open-source oracles. In Figure A.1, we use a small (in comparison with GPT-3.5-turbo) open-source model, Llama2-chat-13B, as the oracle model and generate a batch of poisoned data with the content injection attack. We denote this variant of AutoPoison as `AutoPoison/Llama2-chat-13B` in Figure A.1. We observe a similar advantage of `AutoPoison/Llama2-chat-13B` as `AutoPoison/GPT-3.5-turbo` over the handcrafted baseline.
> This experiment verifies that a small open-source oracle can also effectively achieve the adversary's goal, which further demonstrates the flexibility of the proposed method.

---

> > ### Comment · Reviewer_oPzU · 2023-08-19
> >
> > I thank the authors for their response. I appreciate the evaluation of AutoPoison with an open-source oracle, but without the evaluation of more recent models, I will keep my initial positive score. Wish you good luck with the final decision!

---

### Official Review · Reviewer_9ytD · 2023-07-05

**Soundness:** 2 fair
**Presentation:** 3 good
**Contribution:** 3 good
**Rating:** 6
**Confidence:** 4

**Summary:**

This paper analyzes the threat model for poisoning the data for instruction tuning to steer a language model towards certain behaviors. It proposes a data poisoning pipeline AutoPoison, that uses an oracle LLM to generate adversarial clean labels from perturbed adversarial instructions. AutoPoison is evaluated on two instances - content injection and over refusal - and outperforms the hand-craft baseline in terms of effectiveness and stealthiness.

**Strengths:**

1. Creating a threat model for open-ended text generation and proposing a pipeline for it are novel and necessary in the era of LLMs. Focusing on instruction tuning rather than pre-training is reasonable due to the significantly smaller amount of data required for instruction tuning compared to model pretraining.
2. The analysis of model sizes and data poison ratios is thorough and provides people with a clear idea of how exploitable differently-sized models are under various settings.
3. Although the primary experiments utilize simplistic prompts for two rather basic settings of data poisoning, Section 5 demonstrates the extensibility and potential for the pipeline to be adapted for more specific and fine-grained use cases.

**Weaknesses:**

My concerns about the paper is mostly about the evaluation.
### 1. The evaluation on stealthiness is limited.
The paper's argument for stealthiness is that if the generation quality of a poisoned model is similar to that of a clean model, the poisoning pipeline is stealthy enough. This argument is reasonable to me. However, **I'm not sure perplexity and coherence score is good enough to measure the text quality** under the instruction tuning setting. While they can certainly measure the surface-level fluency of the language, the quality of an instruction-tuned model is more about whether it can faithfully follow the instruction. I think LLM-based metrics might be better in evaluating instruction following ability (such as what they did in Vicuna). I would be more convinced if you used human evaluation or LLM-based evaluation to justify the text quality.
### 2. Other aspects of the model should be considered, for example faithfulness and factuality.
It seems to me that the proposed content injection attack may lead to more hallucinations that contain factual errors. For example, in the third example in Figure 3, the model lists McDonald's as a Swedish company, which is not true. I think these factual errors and other issues caused by hallucination cannot be measured by perplexity and coherence, but they are important for the model's performance. Therefore it may be better to consider a more comprehensive evaluation of the ability of the model.
### 3. Lack of human evaluation
While the examples given look coherent and follow the instruction in a correct way. I can still tell the attacked model from the clean model by spotting the fact that many outputs contain "McDonald's" in a weird way. Although injecting this phrase is the goal of the attack, making it easy for humans to tell can easily leak the adversary's intention. It would be better if you conduct some sort of human evaluation to check if an ordinary user without prior knowledge of the attack can tell an attacked model from a clean one.

Another concern is the limited discussion about defense strategies. I think it would be better to have some preliminary discussion on how the proposed pipeline might be defended.

**Questions:**

1. What are some practical use cases for the attack?
2. Do you think the attack can be stealthy enough so that real humans cannot tell?
3. Considering the fact that instruction datasets are much smaller and often annotated by human beings, do you think the threat model is reasonable? How can an adversary attack the annotation process?

**Limitations:**

The authors have addressed the limitations.

---

> ### Author Rebuttal · Authors · 2023-08-10
>
> > LLM-based evaluation.
>
> Thank you for the constructive feedback. We agree that conventional text quality metrics can be limited in evaluating instruction-tuned models.
> We have adopted your suggestion and conducted LLM-based evaluations on a recent benchmark developed by the Vicuna team: MT-Bench [1].
>
> In Table A.3, we evaluate the poisoned models on MT-Bench. Compared to the clean model, we observe no significant change in the LLM-rated scores among the poisoned ones.
> In Table A.4, we use the same LLM judges to rate the poisoned MT-Bench data generated by the oracle model. We find the content injection attack to have minimal influence on the score, while the over-refusal attack affects the score more prominently. However, note that these poisoned samples will be mixed into a much larger set of clean samples, and the standard deviation suggests that the score varies across clean samples. We think the attack remains stealthy under the LLM-based evaluation.
>
> [1]. Zheng, Lianmin, et al. "Judging LLM-as-a-judge with MT-Bench and Chatbot Arena." arXiv 2023.
>
> > More comprehensive evaluations.
>
> Thank you for the suggestion. We have evaluated the model's factuality on the TruthfulQA benchmark. Results in Table A.1 show little performance degradation on attacked models. The differences in MC1 and MC2 are all within one standard deviation. The results suggest that the proposed attack does not introduce more factual errors to the clean baseline model.
>
> > Human evaluation.
>
> Thank you for this suggestion as well. We agree that a human evaluation would be an even more comprehensive analysis of the proposed method.
> We are working on the IRB review to experiment on human subjects and will include the results in the next version of this paper.
>
> > Practical use cases.
>
> We mentioned some potential adversarial use cases when introducing the two example attacks in Section 3.2. For example, the content injection attack can be used to promote an adversary's affiliations; over-refusal attacks can be adopted by an adversary to compromise a target model (*e.g.*, of their opponents). In addition, there are situations where model owners could deliberately employ the proposed methods, for example, to inject target advertisements into their models (like how apps and websites nowadays have ads for profit).
> Given the flexibility of the poisoning pipeline, we believe it can generate a diverse set of poisoned responses for various potential use cases. We have added these points to our discussion.
>
> > How can an adversary attack the annotation process?
>
> While we think this is always a potential, even for closed-source models, which often rely on outsourced data collection, this issue is immediately noticeable for open-source projects.
> Open-source projects that collect crowd-sourced data, for example, Open-Assist and ShareGPT, are directly at risk, as an adversary could directly participate in the annotation processing by contributing to such projects.
>
> > Discussion on defense strategies.
>
> As briefly mentioned in the abstract and introduction, we hope this work raises awareness of the importance of data quality. We believe a straightforward defense strategy is to improve the data cleaning processing during data collection. For example, implement novel, comprehensive evaluations to filter out compromised samples. We are aware that most data collection processes have quality control, including the aforementioned open-source projects. However, this work reveals a new type of data poison that is hard to detect using conventional and LLM-based metrics.
>
> We want to thank you again for your thoughtful feedback. We hope our additional LLM-based evaluation addresses your comments on stealthiness and our experiments on TruthfulQA answered your question regarding the factuality of attacked models. We would appreciate it if you would consider raising your score in light of our response. We would also appreciate the opportunity to engage further if you have any other questions.

---

> > ### Comment · Reviewer_9ytD · 2023-08-13
> >
> > Thank you for your response!
> >
> > I find it convincing and am raising my score to 6.

---

### Official Review · Reviewer_AQ2w · 2023-07-06

**Soundness:** 3 good
**Presentation:** 3 good
**Contribution:** 3 good
**Rating:** 7
**Confidence:** 4

**Summary:**

This paper proposes AutoPoison, an approach that automatically constructs poisoning data for instruction tuning. AutoPoison replaces training responses with poisoned responses obtained by querying an oracle LM with poisoned instructions. AutoPoison is evaluated on two tasks, content injection and over-refusal attacks. The experimental results show that AutoPoison achieves effective attack while maintaining overall text quality.

**Strengths:**

The paper has the following strengths:
1. The paper is well-written and easy-to-follow. It includes sufficient examples helpful for understanding.
2. The approach is simple (in a good way) and can be generalizable.
3. The evaluation is thorough and clearly demonstrates AutoPoison’s effectiveness.


**Weaknesses:**

### The capabilities of instruction-tuned LMs
A key question that the paper did not answer is if the data poisoning significantly deteriorates the LM’s capabilities. I am aware that text generated by the poisoned model has low perplexity and is coherent with the instruction. But these two metrics do not fully represent the LM’s capabilities. I would suggest evaluating the LM’s capabilities on any standard benchmark, such as HELM and MMLU. If the poisoning does not significantly deteriorate the capabilities, that strengthens the paper’s contribution. Otherwise, the attack is not stealthy, because once the users find out that the LM cannot do some task, they will stop using the LM.

### Other small issues
1. Which oracle model did you use in your experiments? Line 136 only says that the oracle model can be GPT-3.5-turbo.
2. At Line 195, why did you use greedy decoding? Typically text is sampled from LMs, e.g., ChatGPT.
3. For the third example in Table 4, McDonald’s does not have a bold font.


**Questions:**

Please consider addressing the points raised in the “Weakness” section.

**Limitations:**

I believe that the paper provides a sufficient discussion of limitations and potential negative impact.

---

> ### Author Rebuttal · Authors · 2023-08-10
>
> > Evaluation of LM's ability on more comprehensive benchmarks.
>
> Thank you for the constructive feedback. We agree that it is important to maintain the model's ability on general tasks; otherwise, users will stop using it. Therefore, we adopt your suggestion and conduct additional evaluations on the MMLU benchmark.
> We report the results on MMLU in Table A.2. By looking at the averaged accuracy over 57 tasks. We observe no significant performance deterioration in attacked models compared to the clean model. By inspecting the performance on each subtask of MMLU, we find two tasks on which one of the poisoned models (over-refusal attack with AutoPoison) has slightly decreased accuracy.
> (Due to limited time and resources, we chose an objective setting by evaluating the mid-sized models (OPT-1.3B) with the strongest attack (*i.e.*, with the highest poison ratio). We will include the full results in the next version of this paper.)
>
> #### Small issues
> > Oracle models.
>
> Sorry about the confusion. Yes, we use GPT-3.5-turbo as the oracle model for AutoPoison. We have revised the manuscript for clarification. Thank you for catching this.
>
> > Decoding strategy.
>
> Thank you for noticing this detail. We use greedy decoding because it is the decoding strategy adopted by the pre-trained OPT models, as mentioned in [2], and generally accepted as most appropriate for this model family.
>
> [2]. Zhang, Susan, et al. "Opt: Open pre-trained transformer language models." arXiv 2022.
>
> >Format in Table 4.
>
> Thank you for catching this as well. We have updated the font accordingly.
>
>
> Thank you again for the thoughtful feedback. We hope our additional experiments on MMLU and other benchmarks have answered your question regarding the capabilities of instruction-tuned LMs. We would appreciate it if you would consider raising your score in light of our response. We would also appreciate the opportunity to engage further if you have other questions.

---

> > ### Comment · Reviewer_AQ2w · 2023-08-14
> >
> > I have read other reviews and the author rebuttals. I would like to thank the authors for providing the new experiment results, which are convincing. Therefore, I raise my rating from 6 to 7.

---

### Official Review · Reviewer_uhYd · 2023-07-09

**Soundness:** 3 good
**Presentation:** 4 excellent
**Contribution:** 3 good
**Rating:** 7
**Confidence:** 4

**Summary:**

This paper proposed AutoPoison, an automated data poisoning pipeline to showcase two example attacks: content injection and over-refusal attacks over the instruction-tuned models.

Overall, this is a nice paper, I appreciate the authors for tackling this research problem. Authors proposed a sound methodology to model two types of attack. My main criticism is at evaluating the attack success, mainly at the metrics that have chosen to report. I would suggest authors to revisit this section to ground this work.

**Strengths:**

* Timely work that shows eliciting exploitable behaviors from downstream models by injecting poisoned instructions.
* Demonstrate two example attacks with different target behaviors: content injection and over-refusal attacks
* Show that the success rates of the content injection attacks correlate with the scale of the LLM. It actually signifies with the machine-generated data.
> “Intriguingly, we find that larger models, empowered with stronger language modeling and generalization ability, are more susceptible to content injection”
* Nice metric to evaluate the over-refusal attacks via two -staged informative refusal with responses and reasons.

**Weaknesses:**

* Author used machine-generated data for instruction tuning. Given that they use GPT-3.5-turbo as the oracle model, the poisoned responses may also seem to be comparable in fluency. Thus, I don’t think perplexity is a right metric to evaluate the attack's stealthiness.
> “For instruction tuning, we use the English split of GPT-4-LLM, an open-source dataset of machine-generated instruction-following data.”
* It seems like the performance advantage of the hand-crafted baseline might be due to the coherence score calculation. Need better clarification here.
* This is strange. Need explanations.
> “In addition, we observe that under the over-refusal attack, OPT-1.3B, the middle-sized model, learns this behavior the fastest.”

**Questions:**

* Threat model considers the responses to be semantically meaningful and to achieve a qualitative change in model behavior. Are they not two competing objectives?
* Is the over-refusal attack a specialized version of the content injection attack?
> “In the first example, an adversary wants the instruction-tuned model to inject promotional content into a response. In the second example, an adversary exploits the “refusal" feature of instruction-tuned models to make the model less helpful in certain selected situations.”
* What kind of biases inherent to oracle models when generating poison instructions?
* Does this assume that the fluent responses will make it easier in the LLM finetuning process? What about over-fitting the LLM to such examples?
> “Because r_adv is crafted by a language model and not a human, this automated response will already have low entropy according to the language model, making it easy to elevate the likelihood of this response during fine-tuning without a severe change in behavior.”
* Authors argue that poisoned responses are hard to detect manually. Did authors perform any qualitative experiments to support this claim?
> “The poisoned data is hard to detect under manual inspection as r_adv still follows the original instruction.”
* Is the coherence score calculated between the generated response and the gold standard response? I am confused with the following statement.
> “We measure the coherence between the instruction and the model’s response in our setting.”

**Limitations:**

Authors adequately addressed the limitations

---

> ### Author Rebuttal · Authors · 2023-08-10
>
> > More metrics to evaluate the attack's stealthiness.
>
> We agree that perplexity is a limited metric to evaluate the attack's stealthiness. We define stealthiness mainly via text quality, and perplexity is a commonly applicable metric of text quality [1]. We agree that machine-generated texts tend to be lower in perplexity, yet we regard this as a strength of AutoPoison compared to handcrafted poisons, as it makes the poisoned samples less likely to be tossed out by conventional text quality filtering, which commonly relies on perplexity [3].
>
> Inspired by your comment, we designed a new LLM-based evaluation to measure stealthiness. (See global response and Table A.3).
> Compared with the clean model, we observe no significant performance change among the poisoned models. It shows that a stronger LLM-based evaluation cannot distinguish between clean and poisoned models, further validating the proposed attack's stealthiness.
>
> [1]. Li et al. "Contrastive decoding: Open-ended text generation as optimization." ACL 2023.
> [2]. Zheng et al. "Judging LLM-as-a-judge with MT-Bench and Chatbot Arena." arXiv 2023.
> [3]. Wenzek et al. "CCNet: Extracting High Quality Monolingual Datasets from Web Crawl Data" LREC 2020.
>
> > Details about the coherence score and why it is higher in the handcrafted baselines.
>
> We would first like to clarify the details of our coherence score calculation. We follow previous conventions [4] by calculating the cosine similarity between the sentence embeddings of the prefix (i.e., instruction) and the generated texts.
>
> Returning to the question, we find handcrafted poisoned data (in the content injection case) to have a better coherence score. This happens because the handcrafted content injection attack works through minor edits of the golden response. Therefore the changes are minimal, which results in little change in the coherence score. Despite the high coherence score, handcrafted attacks are less effective in achieving the adversary's goals (Figure. 2).
>
> [4]. Gao et al. "SimCSE: Simple Contrastive Learning of Sentence Embeddings" EMNLP 2021.
>
> > OPT-1.3B on the over-refusal attack.
>
> Thank you for bringing up this question. We verified the results and confirmed our observation. Based on the results, we conjecture that an "optimal point" exists in model sizes for the behavior to be learned for complicated target behavior like over-refusal. The mid-sized model can comprehend and learn the target behavior more effectively than smaller models, and compared to larger models, it may also be easier to be overridden by the learned behavior. We think it is an interesting future direction to further investigate this phenomenon.
>
> > How does the threat model make qualitative changes in model behavior while being semantically meaningful?
>
> Since instruction-tuned models are often applied to open-ended questions, many possible answers are all equally semantically meaningful. By making qualitative change, we intend to steer the model toward certain kinds of answers that fit the attack objective. We agree that "qualitative change" may not be an accurate description of the attack objective. We have revised this phrasing in our updated manuscript.
>
> > Is the over-refusal attack a specialized version of the content injection attack?
>
> Although both poison attacks can be realized using the proposed pipeline, we do not deem one to be a specialized version of the other because their goals differ. They exemplify two possible exploitation cases corresponding to two types of data poisoning attacks: model availability attacks (with over-refusal) and model integrity attacks (with content injection).
>
> > What kind of biases inherent to oracle models when generating poison instructions?
>
> If we understand this question correctly, we think it asks what kind of biases we deliberately encourage when crafting poison instructions. We design poison instructions based on the specific attack goal. We elaborated on the motivation for each attack goal in Section 3.2, including the poison instructions we sent to the oracle model. Please let us know if you want to clarify the question. We are happy to engage further.
>
> > Fine-tuning models on examples with low perplexity.
>
> The quoted statement is motivated by our observation that machine-generated poisons are more effective than handcrafted ones. Based on such observation, we assume that low-perplexity (fluent) responses are easier for the model to learn. However, We do not think models are overfitted such training examples, because we test models on a dataset independent of the training data and show that the attacked model can generalize to the test distribution. Below we measure the distribution gap between the training and testing data using MAUVE scores [5].
> |  | train vs. train | test vs. test | train vs. test |
> |--|--|--|--|
> | MAUVE score (&uarr;) | 0.963  | 0.969  | 0.352 |
>
> [5]. Pillutla et al. "Mauve: Measuring the gap between neural text and human text using divergence frontiers." NeurIPS 2021.
>
> > Qualitative experiments to show that poisoned responses are hard to detect manually.
>
> We described the poisoned responses as hard to detect because they are semantically meaningful and instruction-following (supported by conventional and LLM-based evaluations). This is a core objective of our "clean-label" attack scenario. The paper also includes example responses to serve as qualitative results for readers to judge. Nonetheless, we agree that rigorous human evaluations can further support the claim regarding manual inspections. We are working on the IRB review to conduct human evaluations and will include them in the next version of this paper.
>
> Thank you again for your thoughtful feedback. We hope our LLM-based evaluation addressed your comment on the stealthiness metrics. We would appreciate it if you would consider raising your score in light of our response. We would also appreciate the opportunity to provide further information or clarification.

---

> > ### Comment · Reviewer_uhYd · 2023-08-20
> >
> > Thank you for authors addressing my comments on the evaluation. I appreciate the authors who took time for additional experiments, and verified some claims presented in the original paper. Hereby, I raised my original score from 5 to 7 since I am satisfied with the rebuttal.
> >
> > Few additional verifications;
> > * Can authors also present the prompt template used in the LLM evaluation.
> > > We report two sets of scores using GPT-4 and GPT-3.5-turbo as judges

---

> > > ### Author Response · Authors · 2023-08-21
> > >
> > > Thank you for acknowledging our rebuttal and updating the recommendation.
> > >
> > > For the LLM-based evaluation, we use the default prompts provided by MT-Bench [1], which are in their official repository (under the path `fastchat/llm_judge/data/judge_prompts.jsonl`). MT-Bench differs from the original vicuna evaluation in that it uses score-based single-answer gradings, so the prompts we use are those labeled as `"type": single`. We will include the evaluation details in our updated appendix.
> > >
> > >
> > > [1]. Zheng et al. “Judging LLM-as-a-judge with MT-Bench and Chatbot Arena.” arXiv 2023.

---

### Author Rebuttal · Authors · 2023-08-10

### Global response

We thank the reviewers for their constructive feedback. We appreciate the positive comments about our proposed method and the writing of the paper. We acknowledge the potential safety issues raised by the ethics reviewer and have greatly revised and extended the discussion on social impact as recommended.

We have attached a one-page PDF with additional experiments suggested by reviewers:
1. Factuality evaluation of poisoned models on the TruthfulQA benchmark [1] (Table A.1).
2. More comprehensive evaluation of poisoned models on the MMLU benchmark [2] (Table A.2).
3. LLM-based evaluation (MT-Bench [3]) on text quality and poisoned model's instruction-following ability (Table A.3).
4. LLM-based evaluation (MT-Bench [3]) on the text quality of the poisoned examples (Table A.4).
5. The effectiveness of AutoPoison when used with a small open-source oracle model (Llama2-chat-13B [4]) (Figure A.1).

The additional experiments on various benchmarks further support our method by showing that the poisoned models maintain their performance on comprehensive benchmarks and their instruction-following ability, as rated by LLM judges. Further, the result with the open-source oracle shows that the AutoPoison attack remains effective when using a smaller open-source model as the oracle, which demonstrates the flexibility of the proposed method.

As mentioned in our limitation section and suggested by reviewers, we agree that human evaluation can further support the proposed method. However, since this experiment involves human subjects, we are working on gathering the information required for an IRB review. The review process takes more time than what we have for the discussion period, but we will include this additional evaluation in our paper's next version.

In response to the ethics review. We thank reviewer eE8i for their well-thought-out discussion on potential safety issues with our paper. We appreciate their understanding that it is standard practice in the security research community to publish papers identifying novel attacks. With the growing interest in studying the safety aspect of large language models, publishing this work can raise awareness and knowledge about vulnerabilities in modern data pipelines. Because it is better to openly discuss such threats and benchmark their severity in academic forums rather than after they are found in the wild, we believe the benefit outweighs the risk of harm.
We agree that we should expand the discussion on social impact and attend to potential risks, and we have extensively revised this section as recommended.

[1]. Lin et al. "TruthfulQA: Measuring How Models Mimic Human Falsehoods" ACL 2021.
[2]. Hendrycks et al. "Measuring Massive Multitask Language Understanding" ICLR 2021.
[3]. Zheng et al. “Judging LLM-as-a-judge with MT-Bench and Chatbot Arena.” arXiv 2023.
[4]. Touvron et al. "Llama 2: Open Foundation and Fine-Tuned Chat Models" arXiv 2023.

---

### Decision · Program_Chairs · 2023-09-21

**Decision:**

Accept (poster)

**Comment:**

This paper proposed AutoPoison, an automated data poisoning pipeline to automatically create adversarial data for instruction turning. This is an important problem to study for deploying responsible LLM. The paper is well-written and the ideas are novel. The reviewers are generally positive after the rebuttal, which addressed most of the concerns. One remaining one is about only evaluating the OPT models, which are not among the state of the art models. I recommend authors extend their approach to some of the newer models and that would make the paper more complete.